# Identification and characterization of large-scale genomic rearrangements during wheat evolution

**Inbar Bariah, Danielle Keidar-Friedman**[ID]**, Khalil Kashkush**[ID]*

Department of Life Sciences, Ben-Gurion University, Beer-Sheva, Israel

* kashkush@bgu.ac.il

**Data Availability Statement:** All relevant data are within the manuscript and its Supporting Information files.

## Abstract

Following allopolyploidization, nascent polyploid wheat species react with massive genomic rearrangements, including deletion of transposable element-containing sequences. While such massive rearrangements are considered to be a prominent process in wheat genome evolution and speciation, their structure, extent, and underlying mechanisms remain poorly understood. In this study, we retrieved ~3500 insertions of a specific variant of *Fatima*, one of the most dynamic *gypsy* long-terminal repeat retrotransposons in wheat from the recently available high-quality genome drafts of *Triticum aestivum* (bread wheat) and *Triticum turgidum ssp. dicoccoides* or wild emmer, the allotetraploid mother of all modern wheats. The dynamic nature of *Fatima* facilitated the identification of large (i.e., up to ~ 1 million bases) *Fatima*-containing insertions/deletions (indels) upon comparison of bread wheat and wild emmer genomes. We characterized 11 such indels using computer-assisted analysis followed by PCR validation, and found that they might have occurred via unequal intra-strand recombination or double-strand break (DSB) events. Additionally, we observed one case of introgression of novel DNA fragments from an unknown source into the wheat genome. Our data thus indicate that massive large-scale DNA rearrangements might play a prominent role in wheat speciation.

## Background

The evolution of pasta and bread wheats (the *Triticum-Aegilops* group) involved two separate allopolyploidization events. The first occurred ~0.5 MYA and included the hybridization of *Triticum urartu* (donor of the A genome) and a species from section *Sitopsis*, most likely *Aegilops speltoides* (donor of the B genome), leading to the formation of the allotetraploid wild emmer *T. turgidum* ssp. *dicoccoides* (genome AABB) [1–4]. The initial domestication of the wild emmer gave rise to the domesticated emmer wheat *T. turgidum* ssp. *dicoccum* (genome AABB), followed by selection of free-threshing durum wheat (*T. turgidum* ssp. *durum*, genome AABB) [5]. The second allopolyploidization event that occurred ~10,000 years ago included hybridization of the domesticated emmer and *Aegilops tauschii* (donor of the D genome) and led to the generation of the bread wheat *T. aestivum* (genome AABBDD) [3,6].

**Funding:** This study was funded by the Israel Science Foundation - Award Number 322/15, awarded to KK.

**Competing interests:** The authors have declared that no competing interests exist.

Domestication, together with allopolyploidization, is a key event that has shaped the wheat genome through selection [5,7]. Wheat allopolyploids are relatively young species and thus are expected to show limited genetic variation due to the "polyploidy diversity bottleneck". This diversity bottleneck is the result of several factors, namely the short time since allopolyploid formation which is insufficient for the accumulation of mutations, the involvement of only few individuals from the progenitor species in the allopolyploidization event and reproductive isolation of the newly formed allopolyploid from the parental species [8,9]. Nevertheless, wheat allopolyploids show wider morphological variation, occupy a greater diversity of eco-logical niches and proliferate over larger geographical areas, relative to their diploid ancestors [8]. Indeed, the accelerated genome evolution triggered by allopolyploidy may be largely responsible for the wide genetic and morphologic diversity observed in wheat allopolyploids.

Allopolyploidy was shown to trigger a series of revolutionary (i.e., occurring immediately after allopolyploidization) as well evolutionary (i.e., occurring during the life of the allopolyploid species) genomic changes in wheat allopolyploids, which might not be attainable at the diploid level [3,8]. These genomic changes can include the activation of transposable elements (TEs), together with massive and reproducible elimination of TE-containing sequences, as reported for newly formed wheat allopolyploids [10–14]. TEs, corresponding to fragments of DNA able to "move" and proliferate within the host genome, account for over 80% of the wheat genome [5,15–18]. The majority of TEs in wheat allopolyploid genomes are derived from long-terminal repeat retrotransposons (LTRs) that contribute to the highly repetitive nature of those genomes [5,16,18]. Due to their highly repetitive nature, TEs can interact in a disruptive manner during both meiotic recombination and DNA repair processes, leading to a variety of genomic rear-rangements, including sequence translocations, duplications and elimination [11,19–22]. TE activity might trigger DSBs at insertion and excision sites [21,23,24]. Additionally, alternative transposition events can also result in TE-associated chromosomal rearrangements [24].

Large-scale genomic rearrangements between wheat allopolyploids [25–29] and between wheat allopolyploids and their progenitor species [30,31] were previously identified. The mechanism(s) of DNA sequence elimination, including deletion of TE-containing sequences following allopolyploidization events, has yet to be identified. In a previous study, we reported a significant decrease in relative copy numbers of *Fatima* elements in newly formed allohexa-ploids, relative to the expected additive parental copy number [10]. A possible explanation for this result was the rapid elimination of *Fatima*-containing sequences following allopolyploidi-zation events.

In this study, a specific variant of *Fatima*, a well-represented family of *gypsy* LTR retrotran-sposons, was used to identify flanking DNA sequences that had been deleted from wheat allo-polyploid genomes. The use of *Fatima* as a genetic marker, together with the availability of genome drafts for various wheat species, facilitated the identification of large-scale genomic rearrangements between wild emmer and bread wheat. In addition, indel (insertion/deletion) breakpoints were identified and further characterized. Detailed analysis of 11 indels gave rise to possible mechanisms involved in DNA rearrangements following allopolyploidization and/or domestication processes.

## Results

### Utilizing *Fatima* LTR retrotransposons for the identification of large-scale sequence variations between wild emmer and bread wheat

The consensus sequence of the autonomous *Fatima* element (RLG_Tunk_Fatima_consensus-1, 9997 bp in length, solo LTR length 473 bp) was used as a query in a search using MAK soft-ware specifically designed to retrieve *Fatima* insertions based on sequence similarity, together

with their flanking sequences (500 bp from each side), from the draft genomes of wild emmer and bread wheat. Overall, 1,761 full length *Fatima* insertions were retrieved from the wild emmer genome and 1,741 full length *Fatima* insertions were retrieved from the bread wheat genome. The majority of retrieved *Fatima* insertions (97.4% in wild emmer and 97.6% in bread wheat) were located within the B sub-genome (S1 Fig). The remaining retrieved *Fatima* insertions were found in the A sub-genome (36 insertions in wild emmer and 33 insertions in bread wheat), or were unassigned to a specific chromosome (10 insertions in wild emmer and 8 insertions in bread wheat).

The wheat B sub-genome may have undergone massive modifications (yielding the differential genome), as the BB genome donor has yet to be identified and A and D sub-genomes are conserved, a phenomenon referred to as 'pivotal-differential' genome evolution [32]. Thus, the B sub-genome was a promising target in efforts aimed at identifying large-scale genomic rearrangements. In this study, we accordingly focused specifically on chromosomes 3B and 5B. In wild emmer, 268 *Fatima* insertions were retrieved from chromosome 3B and 274 *Fatima* insertions were retrieved from chromosome 5B, while in bread wheat, 274 *Fatima* insertions were retrieved from chromosome 3B and 277 *Fatima* insertions were retrieved from chromosome 5B. Comparative analysis (see Methods part Identification of species-specific *Fatima* insertions) revealed that while the majority of *Fatima* insertions in chromosomes 3B and 5B are common to wild emmer and bread wheat (i.e., monomorphic insertions), 83 (~15%) of the insertions were unique to wild emmer and occurred at polymorphic insertion sites. Several sources for such polymorphism were identified. In 4 of the cases, the presence (i.e., full sites) vs. the absence (i.e., empty sites) of *Fatima* with notable target site duplications (TSDs) were noted. In 47 of the cases, insertions and/or deletions were detected within the *Fatima* element; in some of these instances, the deletion also included part of the *Fatima*-flanking (i.e., chimeric) sequences. Three of *Fatima* insertions were found to be polymorphic due to assembly artifacts resulting in false positive. The remaining 29 *Fatima* polymorphic insertions were found within sequences that were missing from the orthologous loci in bread wheat genome (long indels, ranging in size from 13 kb to 4.4 Mb).

Out of the 29 *Fatima* polymorphic wild emmer-unique insertions, 20 were included within 19 loci in which *Fatima*-containing sequences were replaced by long insertions in bread wheat genome, and 9 were found within sequences that were absent from the orthologous loci in bread wheat genome. The 9 loci together with a single locus out of the 19 previously described loci, showed clear breakpoints and thus were chosen for further analysis. Additionally, a case where two *Fatima* insertions from the wild emmer genome showed high flanking similarity to a single *Fatima* insertion from the bread wheat genome was identified and further analyzed.

Detailed comparative analysis of the above 11 cases were done using a chromosome walking approach and dot plot sequence alignments (S2 Fig) in wild emmer vs. bread wheat genomes. In all 11 cases, indel breakpoints were identified as the borders between high sequence similarity regions (i.e., 95% sequence identity or higher for a word size of 100) to regions that showed low sequence similarity (i.e., lower than 95% sequence identity for a word size of 100) using dot plot representations of the sequence alignments between the orthologous loci in the wild emmer and bread wheat genomes. The lengths of the deleted and/or introduced sequences were defined as the distances between the 5' and the 3' breakpoints. Table 1 summarizes the *in silico* characterization of the 11 loci in wild emmer vs. bread wheat.

## Characterization of large-scale indels borders

To address the underlying mechanisms of large-scale rearrangements, it was important to identify and characterize the indels breakpoints. In 9 of the 11 loci (i.e., $3B_1$, $3B_2$, $3B_3$, $3B_4$, $3B_5$, $5B_1$, $5B_2$, $5B_3$, $5B_4$, Table 1) the indel borders showed sequence homology.

**Table 1. *In silico* characterization of large-sequence variations identified in the bread wheat vs. wild emmer genomes.**

| Locus[1] ID | Location | | Locus length (bp)[4] | | Type of rearrangement |
|---|---|---|---|---|---|
| | Wild emmer[2] | Bread wheat[3] | Wild emmer | Bread wheat | |
| 5B₁ | 5B:566939353–567135082 | 5B:561057394–561064945 | 195730 | 7552 | deletion in bread wheat |
| 3B₁ | 3B:774200469–774452950 | 3B:760803787–760805537 | 252482 | 1751 | deletion in bread wheat |
| 5B₂ | 5B:516702290–516721374 | 5B:511383608–511385023 | 19085 | 1416 | deletion in bread wheat |
| 5B₃ | 5B:363487255–363551431 | 5B:349934346–349934349 | 64177 | 4 | deletion in bread wheat |
| 5B₄ | 5B:587350647–587364130 | 5B:581381548–581381551 | 13484 | 4 | deletion in bread wheat |
| 3B₂ | 3B:284755035–284771490 | 3B:286353814–286353819 | 16456 | 6 | deletion in bread wheat |
| 3B₃ | 3B:493386824–493410158 | 3B:482234389–482234390 | 23335 | 2 | deletion in bread wheat |
| 3B₄ | 3B:538946011–540047920 | 3B:527682008–527682029 | 1101910 | 22 | deletion in bread wheat |
| 3B₅ | 3B:606914695–606946995 | 3B:596314588–596314620 | 32301 | 33 | deletion in bread wheat |
| 5B₅ | 5B:610009239–610050693 | 5B:603942312–603952982 | 41455 | 10671 | introgression of new DNA fragment |
| 5B₆ | 5B:84661892–85585936 | 5B:81624361–82045980 | 924045 | 421620 | copy number variation |

[1] The first number and letter in the locus ID refer to the chromosome in which the genomic locus is found

[2] WEWSeq v1.0 (http://wewseq.wix.com/consortium) coordinates.

[3] IWGSC RefSeq v1.0 (downloaded in June 2017 from: http://plants.ensembl.org/Triticum_aestivum/Info/Index) coordinates.

[4] Locus length was determined as the genetic distance between the 5' and 3' breakpoints/borders of the sequence variation identified using dot plot alignment between the wild emmer and bread wheat genomes (minimum repeat length of 100 bp and 95% repeat identity; S2 Fig). For locus 5B₆, the borders of the repeat units in wild emmer were identified based on dot plot comparison of the locus surrounding locus 5B₆ in wild emmer against itself (minimum repeat length of 100 bp and 95% repeat identity; S3C Fig).

**Indels flanked by long sequence repeats.** In 3 of the 9 loci (3B₁, 5B₁, and 5B₂; Table 1), high nucleotide identity between the 5' and 3' regions of the indel was seen in wild emmer. In the 5B₁ and 5B₂ loci, the sequences absent from bread wheat genome vs. wild emmer were found to contain sequence duplications, with two direct sequence repeats sharing high nucleotide identity (95% or higher) throughout long sequence segments.

Dot plot comparison of the genomic locus surrounding locus 5B₁ from wild emmer chromosome 5B and from bread wheat chromosome 5B revealed a 196 kb sequence from wild emmer genome that lacks long segmental similarity to the orthologous locus in bread wheat (Table 1, S2A Fig). This 196 kb segment borders high sequence similarity regions composed of two direct sequence repeats (S3A Fig) and consists of 71.49% TEs. In bread wheat, the 5B₁ locus is composed of a 7.6 kb segment that shows high nucleotide identity (99%) to both the 5' flanking sequence (nucleotides 1–1355 and 1798- the end of the locus) and the 3' flanking sequence (nucleotides 1–1385 and 1798-the end of the locus) of locus 5B₁ in wild emmer (Fig 1). The 7.6 kb segment from the 5B₁ locus in bread wheat contains three truncated TEs, *Hawi*, *Clifford* and *Conen*, with ~4 kb in the 3' region of the segment being annotated as part of a gene coding for a lipoxygenase. The indel in locus 5B₁ was further validated by PCR analysis using a forward primer based on the 7.6 kb segment in the bread wheat genome, which showed high nucleotide identity to both the 5' and 3' regions flanking the wild emmer 5B₁ locus, and a reverse primer based on the eliminated sequence, which led to wild emmer-specific sequence amplification (Fig 2A). Additional PCR analysis was performed using a forward primer based on the eliminated sequence and a reverse wild emmer-specific primer based on the 3' flanking region of locus 5B₁, which showed high nucleotide identity to the 7.6 kb segment in the bread wheat 5B₁ locus; this also led to wild emmer-specific amplification (Fig 2B). The wild emmer-specific amplification supports bioinformatics-based findings regarding the absence of the 196 kb segment from the bread wheat genome, relative to the wild emmer genome. PCR analysis using the same forward primer as used for the reaction first described in this paragraph and

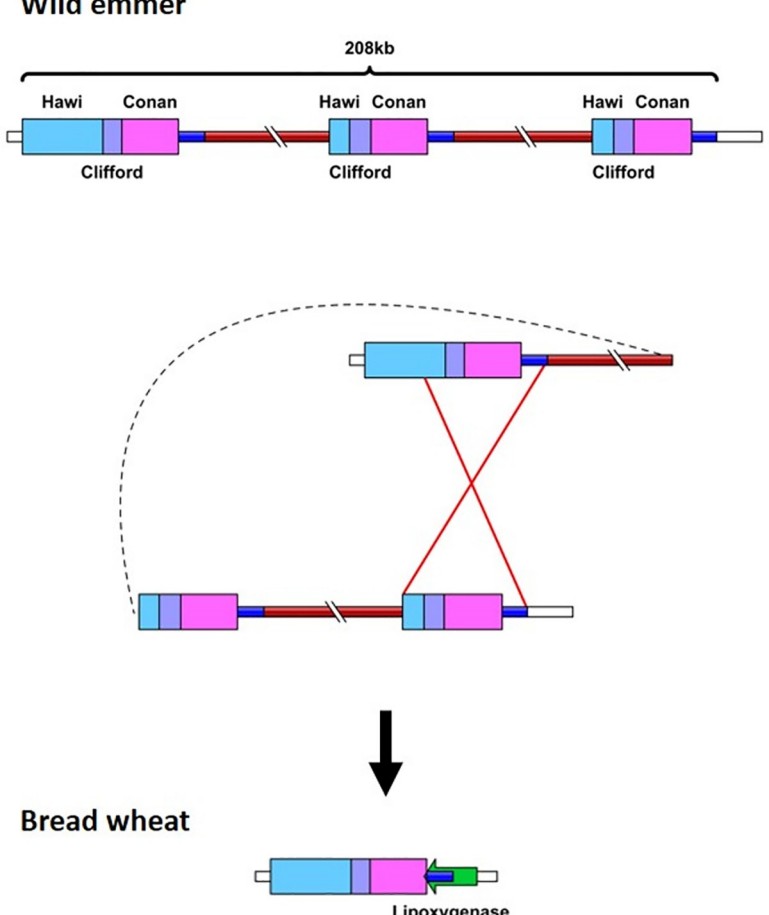

**Fig 1. Schematic representation of the locus containing $5B_1$ in the wild emmer and bread wheat genomes.** Unequal intra-strand recombination involving TEs resulted in a large-scale deletion in bread wheat (bottom) vs. wild emmer (top). Sequence length is unscaled. The *lipoxgenase* gene (TRIAE_CS42_5BL_TGACv1_408003_AA1360930, green arrow) was annotated in bread wheat, while no genes were identified in the orthologous genomic locus in wild emmer. Different colored boxes denote different TE families. Pale blue box notes a retrotransposon. Purple and pink boxes note DNA-transposons. Brown line represents two direct sequence repeats. Blue line represents ~4 kb sequence segment that was annotated as part of a gene coding for a lipoxygenase in bread wheat and was not annotated in wild emmer. Red lines represent the suggested unequal intra-strand recombination event.

the reverse primer based on the indel 3' flanking region led to amplification of wild emmer and bread wheat sequences (Fig 2C), validating the sequence signature identified at the indel borders. The expected PCR products were sequenced for validation (see materials and methods).

A 252 kb sequence from locus $3B_1$ (Table 1) of wild emmer chromosome 3B was not identified on chromosome 3B of bread wheat. However, the orthologous genomic locus was identified in bread wheat based on sequence alignment between the genomic locus containing $3B_1$ from wild emmer and bread wheat chromosome 3B (S2B Fig). The sequence, which was absent from locus $3B_1$ in bread wheat, was composed of two direct sequence repeats (S3B Fig) and consisted of 61.48% TEs. Locus $3B_1$ in bread wheat consisted of a ~1.8 kb segment, which showed 99% nucleotide identity to the sequence found downstream to $3B_1$ in wild emmer. Additionally, a ~1.5 kb fragment in the 3' region of locus $3B_1$ in bread wheat showed 92% nucleotide identity to the sequence found upstream of wild emmer locus $3B_1$. The missing sequence data (Ns) ~1.8 kb

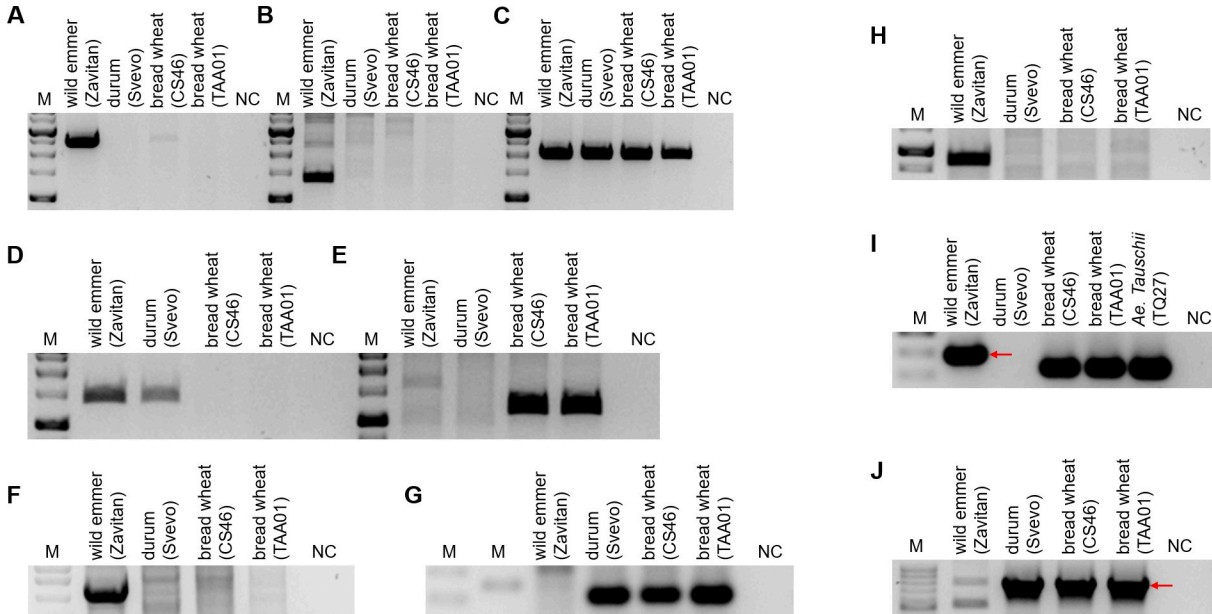

**Fig 2. PCR analysis using primers designed based on Indels identified between wild emmer and bread wheat.** Indel in locus $5B_1$ (A-C): (A) Forward primer designed based on the 7.6 kb segment in bread wheat genome which shows high nucleotide identity to both the 5' region and the 3' region flanking the wild emmer $5B_1$ locus deleted sequence from the wild emmer genome and reverse primer designed based on the deleted sequence. (B) Forward primer designed based on the deleted sequence and reverse wild emmer specific primer designed based on the Indel 3' flanking region of $5B_1$ which shows high nucleotide identity to the 7.6 kb segment in bread wheat $5B_1$ locus. (C) Forward primer as was used for the reaction described for (A) and reverse primer designed based on the Indel 3' flanking. Indel in locus $5B_3$ (D-E): (D) Forward primer designed based on the deleted sequence and reverse primer based on the Indel 3' flanking. (E) Forward primer based on the Indel 5' flanking and the same reverse primer as was used for the reaction described for (D). Indel in locus $3B_4$ (F-G): (F) Forward primer designed based of the 5' flanking sequence of $3B_4$ and reverse primer designed based on the deleted sequence. (G) Forward primer as was used for the reaction described for (F) and reverse primer designed from the 3' flanking of $3B_4$. Indel in locus $5B_5$ (H-J): PCR validations were carried out using primers designed based on the flanking sequences of the Indel coupled with primers designed from the 41 kb wild emmer specific segment (H-I). Expected product length are indicated by red arrows in (I-J). The lower bands in (I) were sequenced and identified in the D sub-genome of the bread wheat. (J) Forward primer designed based on the 11kb bread wheat specific sequence and the reverse primer was the same primer as was used for (I), designed based on sequence located downstream to the Indel. Non-specific amplification was observed for wild emmer. See S1 Table for detailed plant accessions list and S2 Table for primers design and expected products lengths. "M" represents the size marker, "NC" represents for negative control, ddH2O was used as template in PCR reactions. The PCR analysis in (A-C), (D-E), (F), (G), (H), (I) and (J) were visualized on separate agarose gels.

upstream of the 5' breakpoint in the wild emmer genome could have interfered with exact determination of the 5' breakpoint and led to partial alignment of the ~1.8 kb segment to the 5' flanking end of the indel. A truncated *XC* transposable element was identified 10 nucleotides downstream of the 5' end of the 1.8 kb segment in bread wheat and at 10 nucleotides downstream of the locus $3B_1$ 3' end in wild emmer. An additional truncated *XC* transposable element was annotated 1.1 kb upstream of the 5' breakpoint of the indel in locus $3B_1$ in wild emmer.

An additional 19 kb sequence consisting of 99.6% TEs was absent in locus $5B_2$ (Table 1) in bread wheat chromosome 5B, relative to wild emmer. The indel borders were identified using dot plot alignment between the locus containing $5B_2$ in the wild emmer genome and the orthologous locus in bread wheat chromosome 5B. In this manner, the indel breakpoints were determined as the borders of the high sequence similarity regions (S2C Fig) and were both identified within tandem *Inga* LTRs.

## Indels flanked by short sequence repeats

For the remaining 6 loci ($3B_2$, $3B_3$, $3B_4$, $3B_5$, $5B_3$, and $5B_4$) only short sequence identity regions (micro-homology, <10 bp) were identified between the indel borders in wild emmer. In 2 of

the 6 loci ($3B_2$ and $3B_4$) the absent sequence in bread wheat were replaced by short insertion (<15 bp), while in the remaining 4 loci, no novel sequences were introduced to the bread wheat genome instead of the absent sequences. Here described the sequence signatures of the mentioned 6 indels.

A 64 kb sequence consisting of 79.89% TEs in wild emmer chromosome 5B, locus $5B_3$ (Table 1), was absent in the orthologous genomic locus in the bread wheat genome. However, the orthologous locus from which the 64 kb segment was absent was identified in the bread wheat genome based on flanking alignment. Moreover, the indel breakpoints were identified by dot plot comparison of the sequences flanking the $5B_3$ locus in the wild emmer and bread wheat genomes (S2D Fig). Locus $5B_3$ was found to border mononucleotide 'A' at both the 5' and 3' ends in wild emmer, while in bread wheat, the 64 kb segment between the two 'A' mononucleotides was absent. Instead, the 'A' mononucleotide appeared in a single copy between the conserved sequences flanking locus $5B_3$ and both of the 'A' mononucleotides in wild emmer (Fig 3A). The indel 5' breakpoint was identified within the truncated *BARE1* and *WIS* TEs, whereas the 3' breakpoint was identified within a truncated *Fatima* element. PCR analysis using a forward primer based on the deleted sequence and a reverse primer based on the indel 3' flanking region resulted in allotetraploid-specific amplification (Fig 2D). At the same time, PCR amplification using a forward primer based on the indel 5' flanking region and the same reverse primer based on the indel 3' flanking region led to bread wheat-specific amplification (Fig 2E). These results provide additional support for the indel identified in the $5B_3$ locus. The fact that allotetraploid-specific amplification was observed using the forward primer directed against a sequence in locus $5B_3$ which was not identified in the orthologous locus in bread wheat could be explained by the absence of the 64 kb segment from locus $5B_3$ in bread wheat. This would prevent amplification in the examined bread wheat accessions. The observed bread wheat-specific amplification using primers based on the indel flanking sequences suggests that the 64 kb sequence indeed was absent from locus $5B_3$ in bread wheat, resulting in a shorter distance between the surrounding sequences, thus allowing amplification from the bread wheat accessions examined.

In the case of locus $5B_4$ (Table 1), a 13 kb sequence consisting of 81.76% TEs was absent in bread wheat chromosome 5B, as compared to wild emmer. Indel breakpoints were identified by dot plot comparison (S2E Fig), revealing that the 13 kb segment was flanked by the 4-nucleotide motif 'GCGT'. In bread wheat, a single copy of the 'GCGT' motif was identified between

**A**
ATTGCAC**A**    -64kb-    **A**TATGTCT
 ATTGCAC**A**TATGTCT

**B**
AAAGCTA**GCGT**    -13kb-    **GCGT**GGAGGAG
 AAAGCTA**GCGT**GGAGGAG

**C**
A**AATTTG**CAGGAACGGAAACGAATCA**A**    -16kb-    **TTG**GTCCTCA
 A**AATTTG**CAGGAACGGAAACGAATCA**A**aat**TTG**GTCCTCA

**D**
GTCCCCTTCTCTGT    -23kb-    **CCCC**AGATGCT
 GTCCCCTTCTCTGT**CCCC**AGATGCT

**E**
TTTTGATG    -1.1Mb-    **GTC**TTTCCCCTCTCTCTCT**CT**AGCACAACTCCGTC
 TTTTGATGtct    //    **GTC**TTTCCCCTCTCTCTCT**CT**AGCACAACTCCGTC
TTTTGATGtctagcacaactcc**gtc**    //    **GTC**TTTCCCCTCTCTCTCTCT**CT**AGCACAACTCCGTC
 TTTTGATGtctagcacaactcc**GTC**TTTCCCCTCTCTCTCTCT**CT**AGCACAACTCCGTC

**F**
GGATAGG**GA**    -32kb-    TCATCGTCACGTGTTGGGGCGCGCCCTCGT**GA**TC
 GGATAGG**GA**TCATCGTCACGTGTTGGGCGCGCCCCCGC**GA**TC

**Fig 3. Indels result in sequence signatures characterizing DSB repair via MMEJ (A-B) and SD-MMEJ (C-F).** Sequence signatures from genomic loci $5B_3$ (A), $5B_4$ (B), $3B_2$ (C), $3B_3$ (D), $3B_4$ (E), and $3B_5$ (F). The top row represents the indel breakpoints in wild emmer, while the bottom row represents the sequence at the orthologous loci in bread wheat. In (E), the second and third rows represent suggested SD-MMEJ intermediates. Only top strands are shown. Bold-short direct or inverted repeats spanning the DSB which might have been utilized for microhomology during DSB repair. Blue and green- short direct repeats near but not necessarily spanning the DSB that might have been used as primer repeats. Templates used in fill-in synthesis are underlined and net sequence insertions are in lowercase. The length of the deleted sequence is indicated in gray.

the conserved sequences flanking locus 5B₄ and both of the 'GCGT' repeats in wild emmer (Fig 3B). The 5' breakpoint was identified within a *Fatima* element.

A 16 kb sequence consisting of 67.63% TEs from locus 3B₂ (Table 1) in wild emmer chromosome 3B was not identified in bread wheat chromosome 3B. The orthologous genomic locus in the bread wheat genome was identified by alignment of the sequences flanking locus 3B₂ in the wild emmer genome with bread wheat chromosome 3B. The indel breakpoints were identified by dot plot comparison (S2F Fig). The 5' end of the indel was flanked by the mononucleotide 'A', while the 3' end of the indel border was flanked by the trinucleotide 'TTG', which also appeared 22 bp upstream of the 'A' mononucleotide adjacent to the 5' indel breakpoint as part of the sequence 'AAATTTG' (Fig 3C). In the bread wheat genome, the 16 kb segment was absent and a trinucleotide template insertion 'AAT' was identified between the 'A' mononucleotide and the 'TTG' trinucleotide. The indel 5' breakpoint was identified within the truncated TE *Mandrake* and the 3' breakpoint was identified within an intact *Fatima* element.

An additional 23 kb segment from locus 3B₃ (Table 1) in wild emmer consisting of 99.55% TEs was not identified in bread wheat chromosome 3B. However, the orthologous genomic locus from bread wheat was identified by flanking alignment, while the indel breakpoints were determined by dot plot comparison of the locus flanking locus 3B₃ in the wild emmer and bread wheat genomes (S2G Fig). The 5' breakpoint of the indel in locus 3B₃ borders with the dinucleotide 'GT'. Additional 'GT' dinucleotide motif appeared as a tandem repeat 12 bp upstream of the 'GT' dinucleotide adjacent to the 5' breakpoint, followed directly by the 4-nucleotide 'CCCC' motif. The 3' breakpoint of the indel identified of locus 3B₃ was also bordered by a 'CCCC' motif. Finally, the indel 5' breakpoint was identified within a truncated *Xalax* TE and the 3' breakpoint was identified within an intact *Fatima* element.

A 1.1 Mb sequence in the wild emmer 3B₄ locus (Table 1) consisting of 77.64% TEs was found to border mononucleotide 'G' and was not identified within bread wheat chromosome 3B (Fig 3E). However, the orthologous locus was identified in the bread wheat genome based on flanking alignment. The indel breakpoint was identified by dot plot comparison of the genomic region containing the 3B₄ locus in wild emmer chromosome 3B and in bread wheat chromosome 3B (S2H Fig). The indel in locus 3B₄ resulted in a 14 bp insertion into the bread wheat genome ('TCTAGCACAACTCC'), bounded by 'G' mononucleotides, which formed a direct repeat with a sequence found 20 bp downstream of the 'G' mononucleotide adjacent to the 3' breakpoint in wild emmer (Fig 3E). Additionally, a variation in the copy numbers of the dinucleotide 'TC' repeat found 9 bp downstream of the 3' breakpoint was identified in wild emmer (7 tandem repeats of the dinucleotide) and bread wheat (6 tandem repeats of the dinucleotide) was observed. The indel 5' breakpoint was identified within a truncated *Egug* TE. The absent 1.1 Mb sequence included a gene of unknown function and a gene coding for an uncharacterized protein. Additional support for the described indel in locus 3B₄ was obtained upon PCR analysis using primers based on the indel flanking sequences and on the 1.1 Mb sequence identified in the wild emmer genome (Fig 2F and 2G). PCR analysis using a forward primer based on the 5' flanking sequence of locus 3B₄ and a reverse primer based on the 1.1 Mb sequence yielded an emmer-specific amplification (Fig 2F). At the same time, PCR using the same forward primer and a reverse primer designed from the 3' flanking region of locus 3B₄ resulted in amplification in the bread wheat accessions examined but no amplification in wild emmer (Fig 2G).

An additional 32 kb sequence consisting of 98.22% TEs from the 3B₅ locus (Table 1) in wild emmer chromosome 3B was absent in the orthologous locus in the bread wheat genome. The conserved sequences flanking the 3B₅ locus in bread wheat, identified by dot plot alignment (S2I Fig), were found to connected by an apparent blunt end junction. In wild emmer, the 3B₅ locus bordered with the dinucleotide 'GA' at the 5' end and with the dinucleotide 'TC' at the 3'

end. A 4-nucleotide 'GATC' motif was found 29 bp downstream of the dinucleotide 'TC' adjacent to the 3' breakpoint of the indel (Fig 3F). Four SNPs were detected in the sequence found 17–35 bp downstream of the 'TC' dinucleotide adjacent to the 3' breakpoint in wild emmer. The indel breakpoints were found within intact (5' breakpoint) and truncated (3' breakpoint) *Fatima* elements.

**Introduction of DNA fragments of unidentified origin into the wheat genome.** The indel in chromosome 5B locus $5B_5$ (Table 1) was revealed based on sequence alignment of the flanking sequences of a wild emmer-specific *Fatima* insertion into bread wheat chromosome 5B. Following the identification of the orthologous locus in bread wheat chromosome 5B, the indel breakpoints were determined as the borders of the gaps observed in both axes by dot plot comparison of the orthologous loci from the wild emmer and bread wheat genomes (S2J Fig). The indel identified in locus $5B_5$ involved the replacement of a 41 kb segment consisting of 98.18% TEs found in the wild emmer genome with a 11 kb segment consisting of 61.48% TEs located in the orthologous genomic locus in the bread wheat genome (Fig 4). The indel locus $5B_5$ 5' breakpoint was found within a truncated *Karin* TE, while the 3' breakpoint was found within a truncated *Deimos* TE. PCR validation was carried out using primers based on the flanking sequences of the indel coupled with primers designed against the 41 kb wild emmer-specific segment, resulting in wild emmer-specific amplification (Fig 2H and 2I). The third PCR amplification used a forward primer based on the 11 kb bread wheat-specific sequence and the same reverse primer based on sequence located downstream to the indel, as used in the previously described reaction. This third PCR resulted in amplification of both of the examined bread wheat accessions, yet no amplification was observed for wild emmer (Fig 2J). The 11 kb sequence insertion found in locus $5B_5$ in the bread wheat genome was not identified within the wild emmer or *Ae. tauschii* (the donor of the D sub-genome) genomes based on sequence alignment. This phenomenon might be caused by an introgression of a novel sequence into the wheat genome. Introgression of chromosomal segments from alien genomes is known to be facilitated by allopolyploidy in the wheat group [8].

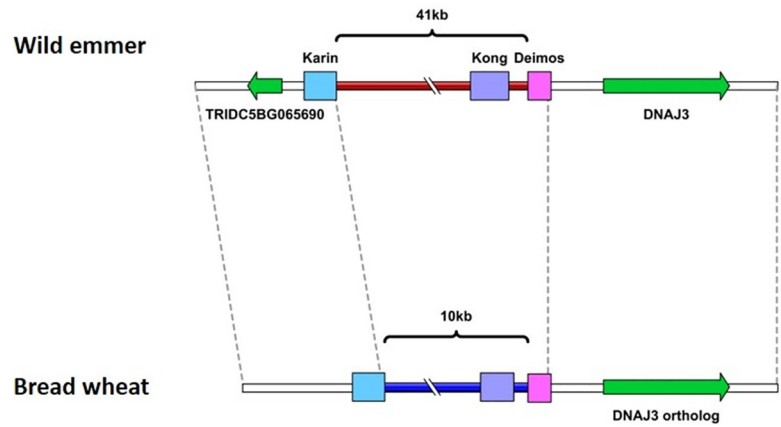

**Fig 4. Schematic representation of locus $5B_5$ in wild emmer (top) and bread wheat (bottom).** Introgression of a new sequence into locus $5B_5$ in the wheat genome. Sequence length is unscaled. Colored boxes denote different TE families. Pale blue box notes retrotransposon. Purple and pink boxes note DNA-transposons. Genes are represented by green arrows. A gene (accession number: TRIDC5BG065690) codes for an undescribed protein and found ~0.5 kb upstream to the *Karin* insertion in wild emmer. A gene (accession number: TRIDC5BG065700) codes for chaperone protein dnaJ3, found ~1 kb downstream from the *Deimos* insertion in wild emmer. A protein coding gene (accession number:: TRIAE_CS42_5BL_TGACv1_405168_AA1321480) in bread wheat shows homology to "TRIDC5BG065700" gene. Brown and blue lines represent the wild emmer and bread wheat specific sequence, respectively. Dashed lines connect between orthologous sequence segments in the borders of the indel and in the ends of the represented sequences.

**Variations in copy numbers of a long tandem repeat in wild emmer vs. bread wheat.**
The analysis of locus $5B_6$ (Table 1) on chromosome 5B revealed variations in the copy numbers of a ~460 kb segment, which appeared as two tandem repeats in the wild emmer genome (totaling 924 kb in length and comprising 79.57% TEs) and in a single copy (422 kb in length and comprising 78.04% TEs) in the bread wheat genome (Fig 5). This copy number variation was identified by dot plot comparison of the orthologous locus surrounding locus $5B_6$ in wild emmer and bread wheat (S2K Fig). The 422 kb segment in locus $5B_6$ in bread wheat showed high sequence similarity (95% or higher with a word size of 100 through long sequence segments) to two repeat units observed in the orthologous locus in wild emmer. The borders of the single repeat unit in bread wheat were determined based on discontinuity points in the sequence alignment (S2K Fig). The borders of the tandem repeats in wild emmer were determined by dot plot comparison of the locus surrounding locus $5B_6$ in wild emmer against itself, as the borders of the regions showing high sequence identity through long sequence segments (95% or higher with a word size of 100) outside of the diagonal line represent the continuous match of the sequence to itself (S3C Fig). In wild emmer, a gene coding for an F-box domain-containing protein was annotated 176 bp downstream of the 5' end of the first repeat, while a gene of unknown function was annotated to the 3' end of the second repeat. Additionally, the first repeat in wild emmer contained a gene coding for the coatomer beta subunit. In bread wheat, the 3' end of the single repeat was identified within a protein coding gene and three additional high confidence protein coding genes were identified within the sequence that underwent copy number variation. The genomic locus in which locus $5B_6$ was found underwent inversion between wild emmer and bread wheat. The borders of the inversion were identified and the inversion length was determined to be ~6.5 Mb (S2L Fig).

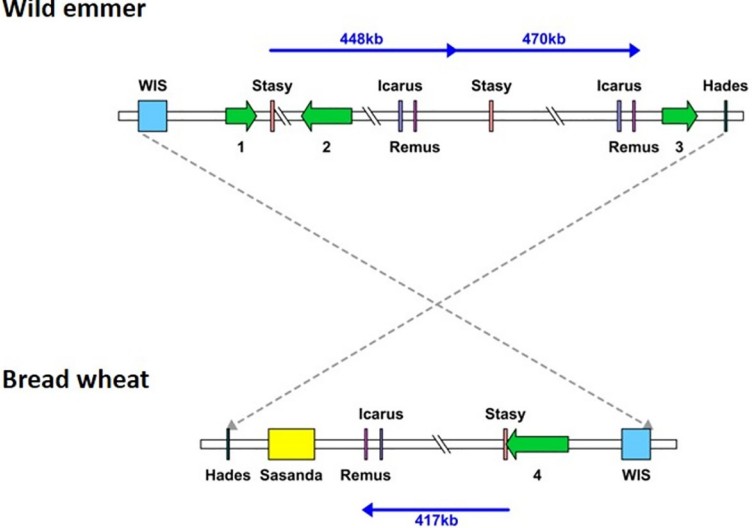

**Fig 5. Schematic representation of locus $5B_6$ in wild emmer (top) and bread wheat (bottom).** Segmental duplication in wild emmer locus $5B_6$. Sequence length is unscaled. Locus $5B_6$ is part of a ~6.5 Mbp segment that underwent inversion between wild emmer and bread wheat. TEs are represented as colored boxes. Pale blue, orange and yellow boxes note retrotransposons. Purple, pink and dark green boxes note DNA-transposons. Genes are denoted by green arrows: (1) F-box domain-containing protein (accession number: TRIDC5BG011160.1); (2) Coatomer, beta subunit (accession number: TRIDC5BG011170.1); (3) Gene encodes for unknown function protein (accession number: TRIDC5BG011180); and (4) Protein coding gene (accession number: TRIAE_CS42_5BS_TGACv1_424303_AA1388580). Dashed lines connect between orthologous in the ends of the represented sequences. The blue line represents the ~460 kb segment, which appears as two tandem repeats in the wild emmer genome and in a single copy in the bread wheat genome.

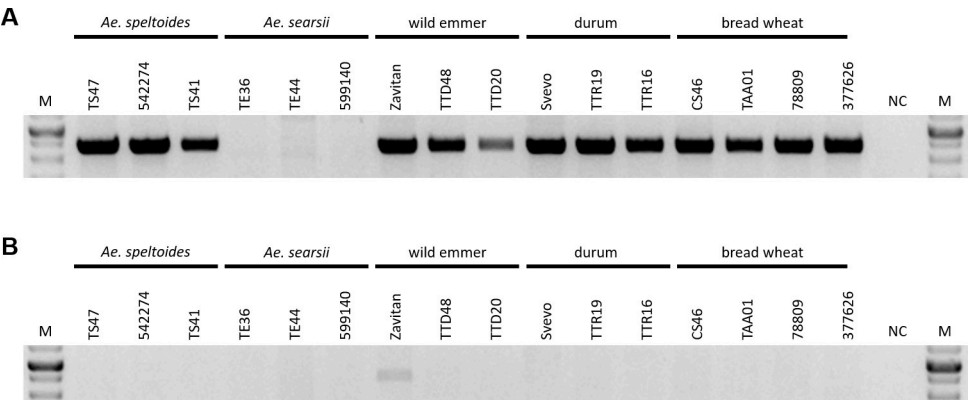

**Fig 6. PCR analysis using primers designed based on copy number variation identified in locus 5B$_6$.** (A) Forward primer designed from the 5' flanking (in wild emmer genome) of the sequence that underwent copy number variation and reverse primer designed from the 5' region of the repeat unit (in wild emmer genome). (B) Forward primer designed based on sequence located in the 3' end (in wild emmer genome) of the segment that underwent copy number variation and the reverse primer that was used for the reaction in (A). "M" represents the size marker, "NC" represents negative control, ddH$_2$0 was used as template in PCR reactions. The PCR analysis was performed for different accessions of wheat allopolyploids (3 wild emmer accessions, 3 durum accessions and 4 bread wheat accessions) and for the available species which are closely related to the diploid B sub-genome donor (3 *Ae. speltoides* accessions and 3 *Ae. searsii* accessions). See S1 Table for detailed plant accessions list and S2 Table for primers design and expected products lengths. The PCR analysis in (A) and (B) were visualized on separate agarose gels.

To identify the origin of the copy number variation and to estimate when this copy number variation emerged, it was important to estimate the numbers of copies of the tandem repeat within different accessions of wheat allopolyploids (3 wild emmer accessions, 3 durum accessions and 4 bread wheat accessions) and within the available species that are closely related to the diploid B sub-genome donor (3 *Aegilops speltoides* accessions and 3 *Aegilops searsii* accessions). The presence of a single repeat was verified by PCR using a forward primer designed against the 5' flanking region (in the wild emmer genome) of the sequence that underwent copy number variation and a reverse primer designed against the 5' region of the repeat unit (in the wild emmer genome). Amplification was observed in all the tested accessions of *Ae. speltoides*, wild emmer, durum and *T. aestivum*, suggesting that the examined segment exists in at least one copy in each of these species (Fig 6A). No amplification was observed for the tested *Ae. searsii* accessions (Fig 6A). To examine whether the ~460 kb segment appears as a tandem repeat in the different accessions, PCR was performed using a forward primer based on the sequence located at the 3' end (in the wild emmer genome) of the segment that underwent copy number variation and the reverse primer that was used in the previously mentioned reaction, resulting in *Zavitan*-specific amplification (Fig 6B). The observed *Zavitan*-specific amplification suggests that the ~460 kb segment is found as a tandem repeat only in this accession, out of the 16 accessions examined. The copy number variation observed in locus 5B$_6$ can be explained by deletion of one of the repeats from the bread wheat genome through unequal intra-strand recombination. Alternatively, the copy number variation seen could be the result of a duplication that occurred within the wild emmer genome later during evolution. The PCR results (Fig 6), together with the high sequence identity between the repeats in wild emmer (S3C Fig), support a scenario whereby the copy number variation is the result of a recent duplication in wild emmer. The boundaries of new segmental duplications in humans were found to be enriched in *Alu*-SINE elements, indicating a possible role for SINE elements in the duplication event [33,34]. The presence of a truncated *Stasy* element (SINE) 2.5 kb downstream of the first repeat start in wild emmer and of a highly similar (99%) truncated *Stasy* element 2.5

kb downstream of the second repeat start could indicate a possible role for this element in the copy number variation reported here.

### Large-scale rearrangements in wild emmer wheat vs. durum

To better assess when the structural variations identified in this study occurred, site-specific PCR analyses were performed for 4 tested accessions, namely a sequenced accession of wild emmer (*Zavitan*), an accession of durum (*Svevo*) and two accessions of bread wheat (accessions CS46 and TAA01). Primers used were based on five of the sequence variations identified in this paper, as described previously (Figs 2 and 6). For the indel of locus $5B_3$, similar amplification patterns from the tested wild emmer and durum accessions (Fig 2D and 2E) suggested this indel occurred following allohexaploidization or during hexaploid wheat evolution. However, for the indels of loci $5B_1$, $3B_4$ and $5B_5$, the similar amplification patterns seen for durum and bread wheat (Fig 2A–2C, 2F, 2G and 2H–2J) indicated that these indels occurred during the evolution of tetraploid wheat, possibly during wheat domestication. The availability of a high-quality durum genome assembly will allow for better characterization of the evolutionary time frame and the events leading to genomic rearrangements in wheat.

## Discussion

DNA rearrangements are known to be prevalent among LTR retrotransposon elements and retrotransposon-containing sequences [11,19,20,35]. In this study, the utilization of *Fatima*, a well-represented *gypsy* LTR retrotransposon family in wheat, as a genetic marker facilitated the identification of such large-scale genomic rearrangements between wild emmer and bread wheat. Detailed analysis of 11 cases of large-scale rearrangements using a chromosome walking approach and dot plot sequence alignments (S2 Fig) of the affected loci in the wild emmer and bread wheat genomes revealed 9 instances of long deletions in bread wheat (5 in chromosome 3B and 4 in chromosome 5B), the introduction of a new DNA fragment, and a single example of copy number variation of a long tandem repeat in chromosome 5B. Detailed analysis of 9 of the 11 loci (i.e., $3B_1$, $3B_2$, $3B_3$, $3B_4$, $3B_5$, $5B_1$, $5B_2$, $5B_3$, $5B_4$, Table 1) led us to suggest two main mechanisms, namely unequal intra-strand recombination and double-strand break repair via non-homologous end-joining (NHEJ).

### Unequal intra-strand recombination

In loci $3B_1$, $5B_1$, and $5B_2$ (Table 1), high nucleotide identity between the 5' and 3' regions of the indel was noted in wild emmer. The TE-containing segments flanking the sequences that were absent in loci $5B_1$ and $3B_1$ in bread wheat vs. wild emmer showed high sequence identity and might have served as a template for unequal intra-strand recombination, resulting in the deletion of the DNA segments between them. Unequal crossing over was recently suggested as being the mechanism involved in the large deletions identified between two allohexaploid wheat cultivars [29]. In the case of locus $5B_2$ both of the indel breakpoints were located within *Inga* LTRs which share the same orientation, suggesting that this rearrangement might have been the result of sequence elimination due to inter-element recombination, as was previously shown in *Arabidopsis* and rice [19,20].

### Double-strand break (DSB) repair via Non-Homologous End-Joining (NHEJ)

For 6 loci ($3B_2$, $3B_3$, $3B_4$, $3B_5$, $5B_3$, and $5B_4$) the indel borders showed only micro-homology (<10 bp), which is not sufficient to serve as a template for homologous recombination [35].

However, the 6 orthologous loci from which the sequences were deleted in the bread wheat genome bear sequence signatures characteristic of DSB repair via NHEJ mechanisms. In eukaryotic cells, DSB repair occurs through two main processes, homologous recombination and NHEJ. In plants, DSB repair occurs more frequently via NHEJ than via homologous recombination [36].

NHEJ pathways for DSB repair can be divided as canonical non-homologous end-joining (C-NHEJ) and microhomology-mediated end-joining (MMEJ) processes [37,38]. The C-NHEJ and MMEJ pathways are template independent-mechanisms and thus can generate a wide range of chromosomal rearrangements, including large deletions and template insertions [36,37,39]. DSB repair via C-NHEJ is favored when end resectioning is blocked, instead relying on the repair of blunt-ended breaks or exploiting small microhomologies during the alignment of broken ends [39,40]. However, when DNA resectioning occurs, other repair pathways, including MMEJ, can compete in repairing the DSB [39]. Therefore, DSB repair via MMEJ generates large deletions more often than does DSB repair via C-NHEJ [38,39].

DNA insertions at the DSB repair site, also known as filler DNA, were previously described in plants [36,41–43]. Filler DNA can be produced when the 3' ends formed at the break site invade a template, such that synthesis is primed based on a short region of homology. Following one or more rounds of template-dependent synthesis, the newly synthesized DNA can join the second end of the DSB, resulting in template insertion [36,38,42,44]. The template for filler DNA synthesis seems more often to be found *in cis*, namely on the same molecule, rather than *in trans*, i.e, on another molecule [36,38,45]. It was proposed that limited DNA synthesis can lead to the presence of microhomology between the DSB ends, which can then be used for DSB repair via synthesis-dependent microhomology-mediated end-joining (SD-MMEJ) [36,38,42,44].

The indels identified in loci $5B_3$ and $5B_4$ (Table 1) were flanked by two short tandem repeats (i.e., 'A' mononucleotides in locus $5B_3$ (Fig 3A) and 'GCGT' motif in locus $5B_4$ (Fig 3B)) in wild emmer, while in bread wheat, the sequence between the short tandem repeats was absent and the repeat unit appeared as a single copy. The sequence signature in bread wheat loci $5B_3$ and $5B_4$ was typical for DSB repair via MMEJ, indicating that the indels in these loci might have resulted from DSB which occurred within the sequences in loci $5B_3$ and $5B_4$ in wild emmer. DSB followed by exonucleases activity and the short tandem repeats that appear in the resulting overhangs could be used for micro-homology in DSB repair via MMEJ. Long deletions with a DSB repair signature similar to that observed in the indels identified in loci $5B_3$ and $5B_4$ were recently described in two allohexaploid wheat cultivars [29].

In the case of locus $3B_2$ a trinucleotide template insertion 'AAT' was identified between the indel borders (the 'A' mononucleotide and the 'TTG' trinucleotide) in the bread wheat genome. This sequence signature in bread wheat might be the result of DSB repair via SD-MMEJ, whereby following DSB within the $3B_2$ locus and end restriction, the 'A' mononucleotide adjacent to the 5' indel breakpoint served as a primer repeat, annealed to the first nucleotide in the complementary strand of the sequence 'AAATTTG' found upstream of the indel 5' breakpoint, thus enabling the synthesis of the 6-nucleotide 'AATTTG'. This synthesis led to trinucleotide ('TTG') micro-homology between the right and left sides of the break, which was used for annealing, and resulted in an indel junction including a trinucleotide insertion ('AAT') and deletion of the 16 kb segment from the $3B_2$ locus (Fig 3C).

Locus $3B_3$ in the bread wheat genome also carries the signature of DSB repair via SD-MMEJ (Fig 3D). The dinucleotide 'GT' might thus have been used as a primer repeat, thereby enabling the synthesis of the 'CCCC' motif (Fig 3D). In this case, the DSB repair via SD-MMEJ resulted in the generation of an apparently blunt repair junction and deletion of the 23 kb segment. Alternatively, the blunt repair junction observed could be the result of DSB repair via

C-NHEJ. However, the long deletion suggests that following the DSB, DNA resectioning based on exonuclease activity occurred. As such, DSB repair via MMEJ is more likely to have occurred [39].

The indel junction in the $3B_4$ locus (Fig 3E) could have arisen as a result of DSB repair that included two rounds of *trans* microhomology annealing and synthesis. In this scenario, during the DSB repair which occurred between the wild emmer and bread wheat genomes, the 'G' mononucleotide found at the 5' end of the DSB served as a primer repeat and annealed to the nucleotide complementary to the 'G' mononucleotide found in the 3' end of the DSB, thus enabling synthesis of the trinucleotide 'TCT'. The newly synthesized 'TCT' motif at the 5' end of the DSB was then annealed to the complementary sequence of the 'TCT' trinucleotide found 20 bp downstream of the 'G' mononucleotide adjacent to the 3' indel breakpoint in wild emmer, thus resulting in the synthesis of the sequence 'AGCACAACTCCGTC'. Following two rounds of nucleotide synthesis, trinucleotide ('GTC') microhomology between the right and left sides of the break used for annealing resulted in an indel junction including a 14 bp templated insertion and the deletion of 1.1 Mb sequence.

The sequence signature in the bread wheat $3B_5$ locus (Fig 3F) corresponded to a site of DSB repair via SD-MMEJ, with the 'GA' motif on the complementary strand to the dinucleotide 'TC' found at the 3' breakpoint serving as a primer repeat used for annealing to the 4-nucleotide 'GATC' motif found 29 bp downstream of the 'TC' dinucleotide adjacent to the 3' breakpoint, thus enabling synthesis of 'TC' dinucleotide on the complementary strand from the 3' end of the DSB. In this scenario, dinucleotide synthesis led to the appearance of dinucleotide ('TC') microhomology between the DSB ends, which were then annealed to yield the apparent blunt end junction seen in the bread wheat genome. The apparent blunt ends junction may also be the result of DSB repair via C-NHEJ. However, repair via C-NHEJ is less likely, considering the length of the deleted sequence.

## Conclusions

In the present study, previous knowledge of how elimination of *Fatima*-containing sequences occurred following allopolyploidization may have contributed to the relative high efficiency of our analysis. Following manual data validation, only 3 (<4%) of the polymorphic insertion sites were removed from the analysis as they were most likely the result of assembly artefacts (missing sequencing data–$N_s$—in one or both of the identified breakpoints). Based on our data, we suggest that sequence deletions mediated through DSB repair and unequal intrastrand recombination, together with the introgression of new DNA sequences, might contribute to the large genetic and morphological diversity seen in wheat allopolyploids and to their ecological success, relative to their diploid ancestors. Such large-scale genomic rearrangements are most likely facilitated by allopolyploidization. The presence of TEs in indels borders suggests a possible role for TEs in the large-scale genomic rearrangements seen in wheat allopolyploids, either by promoting homologous recombination or through other mechanisms. Accordingly, this study aimed to uncover the underlying mechanisms of DNA elimination in wheat, a phenomenon that remained unsolved for many years. Better assembly of the wheat genome drafts will allow for assessing the extent of large-scale DNA rearrangements and evaluating their impact on genome size.

## Materials and methods

### Plant material and DNA isolation

In this study, we used 17 accessions of *Triticum* and *Aegilops* species (S1 Table): 3 wild emmer (*T. turgidum* ssp. *dicoccoides*) accessions, including the sequenced accession *Zavitan*; 3 durum

(*T. turgidum* ssp. *durum*) accessions, including *Svevo*; 4 bread wheat accessions, including two *Chinese Spring* accessions (CS46 and TAA01); six B genome diploid accessions (*Ae. speltoides-* 3 accessions, *Ae. searsii-* 3 accessions) and a single *Ae. tauschii* accession. DNA was extracted from young leaves ~4 weeks post-germination using the DNeasy plant kit (Qiagen, Hilden, Germany).

## Wheat genomic data

The genome drafts of three *Triticum* and *Aegilops* species were used in this study: (1) WEWSeq v1.0 (wild emmer wheat) assembly, a full genome draft of emmer wheat that was sequenced using paired-end and mate-pair shotgun sequencing and assembled using DeNovoMAGIC. The WEW assembly (http://wewseq.wix.com/consortium) contains sorted chromosomes and covers ~95% of the emmer wheat genome [5]. (2) The bread wheat *T. aestivum* Chinese Spring assembly (IWGSC RefSeq v1.0- downloaded in June, 2017 from http://plants.ensembl.org/Triticum_aestivum/Info/Index) was generated by the International Wheat Genome Sequencing Consortium (IWGSC). This assembly covers 14.5 Gbp of the genome with an N50 of 22.8 kbp. Pseudomolecule sequences were assembled by integrating a draft *de novo* whole-genome assembly (WGA), based on Illumina short-read sequences using NRGene deNovoMagic2, with additional layers of genetic, physical, and sequence data [46]. (3) The Aet v4.0 assembly, a reference quality genome sequence for *Ae. tauschii* ssp. *strangulate* (data available from the National Center for Biotechnology Information (NCBI)), was generated using an array of advanced technologies including ordered-clone genome sequencing, whole-genome shotgun sequencing and BioNano optical genome mapping and covers 4.2 Gbp of the genome [47].

## Retrieving *Fatima* insertions from wild emmer and bread wheat draft genomes

A specific variant of intact *Fatima* element and flanking sequences (500 bp from each side) were retrieved from wild emmer and bread wheat draft genomes using MITE analysis kit (MAK) software (http://labs.csb.utoronto.ca/yang/MAK/). MAK is a homology-based software, which allow the use of any TE consensus sequence as query and the BLASTN algorithm with global alignment, to retrieve insertions together with flanking sequences [48,49]. We have previously applied MAK to retrieve *Au* SINE retrotransposon insertion in wheat [14,50]. The publicly available consensus sequence of the *Fatima* element RLG_Null_Fatima_consensus-1 (9997 bp in length) was downloaded from TREP database (http://wheat.pw.usda.gov/ggpages/Repeats/) and used as input (query sequence) in the MAK software. BLASTN was performed against the draft genomes. For retrieval of the *Fatima* sequences, the MAK "member" function was used with an e-value of $e^{-3}$ and an end mismatch tolerance of 20 nucleotides. In addition, flanking sequences (500 bp from each end) were retrieved together with each of the *Fatima* insertions to characterize insertion sites.

## Identification of species-specific *Fatima* insertions

To identify potentially species-specific *Fatima* insertions, the flanking sequences of the retrieved *Fatima* elements from the wild emmer 3B and 5B chromosomes were aligned to the flanking sequences of those elements retrieved from the orthologous chromosomes in bread wheat. Alignments were performed with BLAST+ stand-alone version 2.2.24, using an e-value less than $e^{-100}$. *Fatima* elements in wild emmer for which no flanking similarity was identified in the orthologous bread wheat chromosome were considered as candidate wild emmer-specific insertions and were further examined. Additionally, a case where two *Fatima* insertions

from the wild emmer genome showed high flanking similarity to a single *Fatima* insertion from the bread wheat genome was examined.

## Identification and characterization of *Fatima*-containing sequences that undergo indel and of indel breakpoints

The flanking sequences of the candidate wild emmer-specific insertions were compared to bread wheat chromosome 3B or 5B, depending on the insertion location in wild emmer, using BLAST to identify the orthologous genome locus. In cases where the orthologous genome locus has yet to be identified, a chromosome walking approach was employed, such that longer flanking sequences of the *Fatima* insertion in wild emmer were aligned to the orthologous chromosome from the bread wheat genome using BLAST. Following identification of the orthologous genome locus, dot plot alignments, corresponding to graphical representations of sequence aliments, were performed on orthologous loci to identify sequence variations, using UGENE version 1.23.0 [51] with a minimum repeat length of 100 bp and 95% repeat identity. For each indel observed, the sequence alignments were analyzed and the breakpoints, namely regions where sequence similarity broke down, were identified. To determine indel lengths, the distance between two breakpoints was calculated, based on a minimum repeat length of 100 bp and 95% repeat identity.

To further characterize indels, breakpoints and deleted and inserted sequences were annotated to genes and TEs. Gene annotation was performed using The Grain-Genes Genome Browsers (https://wheat.pw.usda.gov/GG3/genome_browser) for wild emmer (WEWSeq v.1.0) and bread wheat (TGACv1) and the *EnsemblPlants* (http://plants.ensembl.org/Triticum_aestivum/Info/Index) genome browser for bread wheat (TGACv1). TE annotation was performed using Repeat-Masker (http://www.repeatmasker.org/) with a cutoff of 250 and TE databases of wheat transposable elements taken from TREP (http://wheat.pw.usda.gov/ggpages/Repeats/) from the nrTREP database for *Triticum* and *Aegilops* species. Finally, schematic presentations (see Figs 1, 4 and 5) were prepared using IBS version 1.0.3 (http://ibs.biocuckoo.org/index.php) [52].

## PCR analysis

PCR validation was performed using primers designed with PRIMER3 version 4.1.0 based on identified sequence variations (see S2 Table for primer sequences), such as deleted or newly introduced sequences and sequences flanking deletions. To generate PCR products up to 800 bp, each reaction contained: 10μl PCRBIO HS Taq Mix Red (PCRBiosystems), 7 μl ultrapure water (Biological Industries), 1 μl of each site-specific primer (10μM) and 1 μl of template genomic DNA (approximately 50 ng/μl). The PCR conditions were 95˚C for 2 min, 35 cycles of 95˚C for 10 sec, the calculated annealing temperature for 15 sec and 72˚C for 15 sec. For PCR products longer than 800 bp, each reaction contained 12 μl ultrapure water (Biological Industries), 4 μl of 5X PrimeSTAR GXL Buffer (TaKaRa), 1.6 μl of 2.5 mM dNTPs, 0.5 μl of each site-specific primer (10 μM) and 0.4 μl of PrimeSTAR GXL DNA Polymerase (1.25 U, TaKaRa). The PCR conditions used were 94˚C for 5 min, 30 cycles of 98˚C for 10sec, the calculated annealing temperature for 15 sec and 68˚C for 1 min. PCR products were visualized in 0.8–1% agarose gels. Note that expected PCR products were extracted from the agarose gel, and sequenced for validation. Figures were prepared using GIMP (https://www.gimp.org/), and Microsoft PowerPoint.

## Supporting information

**S1 Fig.** Distribution of *Fatima* specific variant in chromosomes 1–7 of sub-genome A (blue) and sub-genome B (pink) in wild emmer (A) and bread wheat (B). Unmapped *Fatima* insertions are not shown.
(PDF)

**S2 Fig.** Dot plot representations of genomic loci containing sequence variations identified between wild emmer (x axis) and bread wheat (y axis) in loci $5B_1$ (A) $3B_1$ (B) $5B_2$ (C) $5B_3$ (D) $5B_4$ (E) $3B_2$ (F) $3B_3$ (G) $3B_4$ (H) $3B_5$ (I) $5B_5$ (J) $5B_6$ (K) and a ~6.5 Mb inversion including the genomic locus $5B_6$ (L). The parameters for the sequence alignments were minimum repeat length of 100 bp and 95% repeats identity. Green- direct repeats, red- inverted repeats. Indels break points\ borders are indicated by black arrows. The numbers in brackets refer to the coordinates of the selected sequences in the WEWSeq_v.1.0 assembly (for wild emmer) and in the IWGSC assembly (for bread wheat).
(PDF)

**S3 Fig. Dot plot representations of genomic loci containing segmental duplication in wild emmer genome.** In $5B_1$ (A) and $3B_1$ (B) loci, the deleted sequences and the Indels borders were found to contain sequence duplications. (C) Recent sequence duplication in locus $5B_6$ identified in wild emmer relative to bread wheat. The parameters for the sequence alignments of the genomic loci against themselves were minimum repeat length of 100bp and 95% repeats identity. Green- direct repeats, red- inverted repeats. Indels break points\ borders are indicated by black arrows. The numbers in brackets refers to the coordinates of the selected sequences in the WEWSeq_v.1.0 assembly.
(PDF)

**S1 Table. Plant accessions used in PCR analyses.**
(PDF)

**S2 Table. Primer used for PCR analyses.**
(PDF)

**S1 Raw images.**
(DOCX)

## Acknowledgments

We want to thank Dr. Guojun Yang, University of Toronto, for providing the stand-alone version of MAK. We also thank Prof. Avi A. Levy, Weizmann Institute of Science, and Prof. Assaf Distelfeld, Tel-Aviv University, for their critical reading of the manuscript.

## Author Contributions

**Data curation:** Inbar Bariah, Danielle Keidar-Friedman.

**Formal analysis:** Inbar Bariah, Danielle Keidar-Friedman.

**Funding acquisition:** Khalil Kashkush.

**Investigation:** Danielle Keidar-Friedman.

**Software:** Inbar Bariah.

**Supervision:** Khalil Kashkush.

**Validation:** Inbar Bariah, Danielle Keidar-Friedman.

**Visualization:** Inbar Bariah.

**Writing – original draft:** Inbar Bariah, Danielle Keidar-Friedman, Khalil Kashkush.

**Writing – review & editing:** Khalil Kashkush.

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
