## [Decision Letter · Decision Letter 0]

28 Oct 2019

PONE-D-19-26784

Identification of large-scale genomic rearrangements during wheat evolution and the underlying mechanisms

PLOS ONE

Dear Professor Kashkush,

Thank you for submitting your manuscript to PLOS ONE. After careful consideration, we feel that it has merit but does not fully meet PLOS ONE’s publication criteria as it currently stands. Therefore, we invite you to submit a revised version of the manuscript that addresses the points raised during the review process.

ACADEMIC EDITOR: Please insert comments here and delete this placeholder text when finished. Be sure to:

Indicate which changes are required versus recommended for acceptanceRequired Changes: As indicated by the AE below, the paper must be written so that the data is not summarized in context of the models. Rather than data should be more fully shown and used to derive the models. Mixing the two together is misleading and illogical. The Supplementary data are also required within the Figures section and sequences should be provided in Figures when mentioned in the text.The hypotheses are speculative. The speculative nature of the hypotheses need to be clearly described and should not be presented as conclusions in  the title. What was the basis of the choice of the 11 cases investigation? The indels require further sequencing to be verified. Place Figures S4 to S8) within the text.Replace accession numbers with species names in the Supplementary data 4-8.The methods require more clarity as described by the two Reviewers. Why was MAK software used?Clarify the use of Fatima as a marker for overall rearrangement.The Stacy involvement is unconvincing. Please modify to accommodate Reviewer 2's critique.There are many textually vague or poorly described terms and logic (spelling, intact vs non-intact elimination versus deletion, etc, as well as evidence that Fatima activation is a late step). Please modify as suggested by Reviewers.Address any conflicts between the reviewsNo significant conflicts were present between the Reviewers.Provide specific feedback from your evaluation of the manuscriptAs noted above, the paper must be rewritten substantially so that the data is not summarized in context of the models. Rather than data should be more fully shown and used to derive the models (that is a bottom up logic). Mixing the two together is misleading, illogical, unreviewable, and inaccessible to the general reader. The Supplementary data are also required within the Figures section and sequences should be provided in Figures when mentioned in the text. 

We would appreciate receiving your revised manuscript by Dec 12 2019 11:59PM. To enhance the reproducibility of your results, we recommend that if applicable you deposit your laboratory protocols in protocols.io, where a protocol can be assigned its own identifier (DOI) such that it can be cited independently in the future. For instructions see: http://journals.plos.org/plosone/s/submission-guidelines#loc-laboratory-protocols

We look forward to receiving your revised manuscript.

Kind regards,

Arthur J. Lustig, PhD

Academic Editor

PLOS ONE

Journal Requirements:

Please provide an amended Funding Statement that declares *all* the funding or sources of support received during this specific study (whether external or internal to your organization) as detailed online in our guide for authors at http://journals.plos.org/plosone/s/submit-now.  Please state what role the funders took in the study.  If any authors received a salary from any of your funders, please state which authors and which funder. If the funders had no role, please state: "The funders had no role in study design, data collection and analysis, decision to publish, or preparation of the manuscript."

Reviewers' comments:

Reviewer's Responses to Questions

**Comments to the Author**

1. Is the manuscript technically sound, and do the data support the conclusions?

Reviewer #1: Partly

Reviewer #2: Partly

2. Has the statistical analysis been performed appropriately and rigorously? 

Reviewer #1: N/A

Reviewer #2: N/A

3. Have the authors made all data underlying the findings in their manuscript fully available?

Reviewer #1: No

Reviewer #2: Yes

4. Is the manuscript presented in an intelligible fashion and written in standard English?

Reviewer #1: Yes

Reviewer #2: Yes

5. Review Comments to the Author

Reviewer #1: In their paper Bariah I. and co-authors describe a list of eleven large indels associated with a transposable element (Fatima) and that segregate presence-absence variation between bread wheat and wild emmer species. The structural rearrangements are thoroughly described and followed by hypotheses about the molecular mechanisms that originated those rearrangements. Despite being explained and referred, the hypotheses remain mostly speculative, which is not always clear in the text and part of the title of the current manuscript. The eleven indels described in the manuscript were validated by PCR. However, without further sequencing of the amplified fragments, neither a molecular marker scale for the resulting electrophoresis gels (Figures S4 to S8), the validation remains unclear.

Additional notes:

The main figures (figure 1 to 4) simplify well the long description of the given rearrangements but I think the “real” alignments would fit in the supplementary data as well or a figure supporting the “high nucleotide identity” at those sequences.

In the text, to which subfigure (A-B-C-D) it refers is missing.

In figures S4 to S8 I would mention the species and not the accession for clarity

In methods, the genome assemblies and annotation (gene and TE) used in the study are unclear, I would refer to the accession numbers.

Reviewer #2: The authors conducted comparative analyses of Triticum aestivum (bread wheat) and Triticum turgidum ssp. Dicoccoides (wild emmer) Fatima transposable elements and flanking sequences. The authors described some results showing that some insertions/deletions (indels) occurred via unequal intra-strand recombination or double-strand break (DSB) events, and that a number of these events occurred at or near Fatima and other TEs.

The main conclusion of the paper is that massive large-scale DNA rearrangements induced by transposons played a prominent role in wheat speciation. The authors show that a number of rearrangement breakpoints are located within Fatima transposons; however, this result is expected based on the method by which these cases were selected for study. In addition, other breakpoints are located in non-Fatima TEs. Thus it is difficult from the data presented to assess the true proportion of TE-induced rearrangements, and whether this figure is greater than would be expected by random chance considering that 80% of the wheat genome is composed of transposon sequences.

Comments and questions:

Line 100, the authors stated that “The consensus sequence of the autonomous Fatima element was used as a query”. Additional description of the query sequence is given in Materials and Methods, but some of this information should be presented in the paper main text (the length of the complete element and LTRs). The authors should indicate what criteria are used to distinguish “intact” from “non-intact” elements. Finally, it would be of some interest to indicate the number of solo LTRs derived from full-length Fatima.

Line 101, “MAK software designed to retrieve Fatima insertions”, the authors should explain why this software is used in this study. Based on the information from the software website, MAK stands for MITE Analysis Kit, which is designed to facilitate automated analysis of miniature inverted repeat transposable elements (MITEs). While Fatima is a retrotransposon, the authors should describe the reason that using this MITE tool to analysis a retrotransposon.

Line 114, Indicate why these particular chromosomes were chosen for analysis? Why is the location of Ph1 on 5B important to this study?

Line 123-129. Authors should replace percentages (“5%, 57%, 34% and 4% of the cases”) with actual numbers.

Line 134: How were these 11 cases chosen for study?

Line 147: Authors state that “Fatima was most likely activated following allohexaploidization”; the reasoning here is a little murky. Possibly could be helped by summarizing number of copies eliminated. Also, is there any evidence to show that the transposon was inactive before the allohexaploidization?

Line 251: DBS, should be DSB

Line 347: templet; should be template?

Line 411, 412: regarding the proposed introgression of an 11 kb segment: Can the TEs contained within this segment indicate its probable origin?

Lines 467-469: The potential role of Stasy elements in the segmental duplication is overstated. The first Stasy is 2.5 kb from the duplication endpoint, and the second is indicated as “in the 5’ region” of a 470 kb duplication. As depicted in Figure 4, this second Stasy may be 50 kb or more from the duplication endpoint. If these elements were involved in the duplication events, one would expect them to be present at or very close to the breakpoints.

Line 772: bends, should be bands

Line 775: revers, should be reverse

In this paper, do ‘elimination’ and ‘deletion’ carry the same meaning?

Some loci described in the study have “high” TE content (Lines 186, 214, 230, 263). Considering that wheat genome is already 80% TE sequences, what does “high” mean?

It is recommended that Supplemental Figure 7 be included in the manuscript as a main text figure. In fact, Supplemental Figures 4, 5, 6, and 7 could all be combined into a single figure for the main text. These PCR results don’t take much space, and it would be helpful for these data to be readily available to the reader.

6. PLOS authors have the option to publish the peer review history of their article (what does this mean?). If published, this will include your full peer review and any attached files.

Reviewer #1: No

Reviewer #2: No

---

## [Author Response · Author response to Decision Letter 0]

17 Jan 2020

Dear Editor,

We would like to thank you and thank the reviewers for your constructive comments. Your comments allowed us to improve the data presentation and the quality of the manuscript. We have agreed to all of the comments and performed the requested changes including repeating PCR reactions for sequence validation. We provide below a point by point response to all comments. Note that page and line numbers are based on the unmarked (without track changes) version of the manuscript. 

Editor comments:

1) As indicated by the AE below, the paper must be written so that the data is not summarized in context of the models. Rather than data should be more fully shown and used to derive the models. Mixing the two together is misleading and illogical. The Supplementary data are also required within the Figures section and sequences should be provided in Figures when mentioned in the text. 

Response: We have made text rearrangements and separated the results and discussion sections. We also have changed the representation style of the data. Furthermore, we have included supplemental figures to the main text as suggested by the reviewers. For example; Figs. S4- 8 are now represented as Figs 2 and 6 in the text. Sequence coordinates are found in Table 1 and sequence alignments are represented as dot plots found in S2-3 Figs. Sequence comparisons and relevant annotations are found in Fig 1 and Figs. 3- 5. See below our response to reviewers’ comments. 

2) The hypotheses are speculative. The speculative nature of the hypotheses need to be clearly described and should not be presented as conclusions in the title. 

Response: Done. We have separated the results and discussion sections, and modified the manuscript title and some of the sections titles. 

a. What was the basis of the choice of the 11 cases investigation? 

Response: See text (lines 128-135). Out of the 29 Fatima polymorphic wild emmer-unique insertions, 20 were included within 19 loci in which Fatima-containing sequences were replaced by long insertions in bread wheat genome, and 9 were found within sequences that were absent from the orthologous loci in bread wheat genome. The 9 loci together with a single locus out of the 19 previously described loci, showed clear breakpoints and were chosen for further analysis. Additionally, a case where two Fatima insertions from the wild emmer genome showed high flanking similarity to a single Fatima insertion from the bread wheat genome was identified and further analyzed.

Detailed comparative analysis of the above 11 cases were done using a chromosome walking approach and dot plot sequence alignments (S2 Fig) in wild emmer vs. bread wheat genomes. 

b. The indels require further sequencing to be verified. 

Response: We routinely do this kind of validation for PCR products. Here as your and reviewer request, we have repeated PCR reactions, and sequenced the products and validated them by sequencing. We have mentioned this in the manuscript (in material and methods and results sections (for example see lines 619-622). 

c. Place Figures S4 to S8) within the text. 

Response: Done, they are now Figs 2 and 6

d. Replace accession numbers with species names in the Supplementary data 4-8. 

Response: Done, see Figs 2 and 6.

e. The methods require more clarity as described by the two Reviewers. Why was MAK software used? 

Response: Done. See below our response to Reviewer 2.

f. Clarify the use of Fatima as a marker for overall rearrangement. 

Response: See below our response to Reviewer 2.

g. The Stacy involvement is unconvincing. Please modify to accommodate Reviewer 2's critique.

 Response: Done. See below our response to Reviewer 2.

h. There are many textually vague or poorly described terms and logic (spelling, intact vs non-intact elimination versus deletion, etc, as well as evidence that Fatima activation is a late step). Please modify as suggested by Reviewers. 

Response: Done. See details below 

 Reviewer #1:

1) In their paper Bariah I. and co-authors describe a list of eleven large indels associated with a transposable element (Fatima) and that segregate presence-absence variation between bread wheat and wild emmer species. The structural rearrangements are thoroughly described and followed by hypotheses about the molecular mechanisms that originated those rearrangements. Despite being explained and referred, the hypotheses remain mostly speculative, which is not always clear in the text and part of the title of the current manuscript. The eleven indels described in the manuscript were validated by PCR. However, without further sequencing of the amplified fragments, neither a molecular marker scale for the resulting electrophoresis gels (Figures S4 to S8), the validation remains unclear. 

Response: The main title and the subtitles of the manuscript were changed. We also have separated the results and the discussion sections. In addition, as requested, unique PCR bands were verified by sequencing and DNA ladders were used to estimate product length (see above response to editor comment). Note that from our experience, no need for such validation to be included in the supplementary. But, if you request that, we can add Fasta format sequences. For example: 

>5B5_zavitan_expected_391bp_product

CGCCGGTTAGTAAAAGCACCTGTTTCTCTCAAAAAAAAAAGTAAAAGCACCTGTAACTGACGCTCAGGGCGCCAAATAGGATTCGCCCGCACCCGCGCCCGCACCGCAACAGCATCCAGGTCGCCAGGTCGGTCCTCTAACCGGAGTCAAACTCAACCACGCGCTCCCCTCTGCGCCGAACCCTCCTCCCCCTTTCGCCAGAACACCCCGGAAGCATCTGGAAGCCCGGCCCTCAGCCCCCGCCACCGCCCGATCCGGATCCGACGGTCCGCGCCCGCCTCTCCCCCGCCCTCCCGCATCTTCGAGAGCGTTCACTCCCGACCACCCGTCCCGCGCTCCGCTATAACTACGACCGCCCCCTCTCCCCCACCTCTCCTCACAAACCGATCCC

>Forward_primer_5B5_zavitan_391bp_product

TGTTYYAAAAAAARTAAAAGCACCTGTAACTGACGCTCAGGGCGCCAAATAGGATTCGCCCGCACCCGSGCCCGCACCGMAACAGCATCCAGGTCGCCAGGTCGGYCCTYTAACCGGAGTCAAACTCAACCACSCSCTCCCCTYTGSGCCGAACCCTCCTCCCCCTTTCGCCARAACACCCCGGAAGCATYKGGAAGCCCGGCCCTMAGCCCCCGCCACCGCCCGATCCGGATCCRACGGTCCGSGCCCGCYTYTCCCCCGCCYTCCCGMATYTTCRARAGCGTTMACTCCCGACCACCCGTCCCGSGCTCCGYTWTAACTACRACCGCCCCCTYTCCCCCACCTYTCCTMACAAACCGATCCMA

>Reverse_primer_5B5_zavitan_391bp_product

TTGTGGGGGGGGCGGTCGTAGTTATAGCGGRCGCGGGACGGGTGGTCGGGAGTGAACGCTCTCGAAGATGCGGGAGGGCGGGGGAGAGGCGGGCGCGGACCGTCGGATCCGGATCGGGCGGTGGCGGGGGCTGAGGGCCGGGCTTCCAGATGCTTCCGGGGTGTTCTGGCGAAAGGGGGAGGAGGGTTCGGCGCAGAGGGGAGCGCGTGGTTGAGTTTGACTCCGGTTAGAGGACCGACCTGGCGACCTGGATGCTGTTGCGGTGCGGGCGCGGGTGCGGGCGAATCCTATTTGGCGCCCTGAGCGTCAGTTACAGGTGCTTTTACTTTTTTTTTTRARARAAMCAGGKGYTTTWMYAAMCCCGGSGA

2) The main figures (figure 1 to 4) simplify well the long description of the given rearrangements but I think the “real” alignments would fit in the supplementary data as well or a figure supporting the “high nucleotide identity” at those sequences. 

Response: Sequence coordinates are found in Table 1 and the genome assemblies used here are publicly available. Sequence alignments are represented as dot plots found in S2-3 Figs. Since some of the loci are over 1 Mb in length, we avoided full sequence alignments. Sequence comparisons and relevant annotations are found in Figs 1 and 3-5. 

3) In the text, to which subfigure (A-B-C-D) it refers is missing. 

Response: Fixed

4) In figures S4 to S8 I would mention the species and not the accession for clarity 

Response: Done, they are now Figs 2 and 6

5) In methods, the genome assemblies and annotation (gene and TE) used in the study are unclear, I would refer to the accession numbers. 

Response: Done under " Wheat genomic data" and "Identification and characterization of Fatima-containing sequences that undergo indel and of indel breakpoints".

Reviewer #2: 

1) The authors conducted comparative analyses of Triticum aestivum (bread wheat) and Triticum turgidum ssp. Dicoccoides (wild emmer) Fatima transposable elements and flanking sequences. The authors described some results showing that some insertions/deletions (indels) occurred via unequal intra-strand recombination or double-strand break (DSB) events, and that a number of these events occurred at or near Fatima and other TEs.

The main conclusion of the paper is that massive large-scale DNA rearrangements induced by transposons played a prominent role in wheat speciation. The authors show that a number of rearrangement breakpoints are located within Fatima transposons; however, this result is expected based on the method by which these cases were selected for study. In addition, other breakpoints are located in non-Fatima TEs. Thus it is difficult from the data presented to assess the true proportion of TE-induced rearrangements, and whether this figure is greater than would be expected by random chance considering that 80% of the wheat genome is composed of transposon sequences. 

Response: In this work, Fatima were used as a genetic marker here and was chosen since it was showed to underwent elimination in the first four generations of a newly formed wheat allohexaploid. It was not claimed that Fatima elements were involved in the mechanisms leading for the 11 indels. TEs might have a role in the rearrangements, i.e., as sequence repeats enabling unequal intra-strand recombination. For clarification, the line discussing the TEs abundant in indels break points was removed from the abstract and the roll of Fatima as genetic marker was further discussed in the text. 

2) Line 100, the authors stated that “The consensus sequence of the autonomous Fatima element was used as a query”. Additional description of the query sequence is given in Materials and Methods, but some of this information should be presented in the paper main text (the length of the complete element and LTRs). 

Response: Done (lines 97-98)

3)The authors should indicate what criteria are used to distinguish “intact” from “non-intact” elements. 

Response: The terminology was corrected, the retrieved elements are not necessarily intact, but full-length elements (line 101) 

 4) Finally, it would be of some interest to indicate the number of solo LTRs derived from full-length Fatima. 

Response: Since Fatima was used as a genetic marker, Fatima content in the genome was not the focus of this study. Fatima content and distribution in the bread wheat genome was recently described in a paper [1] (see references). 

5) Line 101, “MAK software designed to retrieve Fatima insertions”, the authors should explain why this software is used in this study. Based on the information from the software website, MAK stands for MITE Analysis Kit, which is designed to facilitate automated analysis of miniature inverted repeat transposable elements (MITEs). While Fatima is a retrotransposon, the authors should describe the reason that using this MITE tool to analysis a retrotransposon.

 Response: Since Fatima was used as a genetic marker, it was efficient to retrieve Fatima insertions base one sequence similarity. MAK enables the retrieval of TEs based on sequence similarity and specifically similarity of the terminal regions of the sequence. Clarification is added to the text (line 99). For this reason, the retrieved elements were defined as a specific variant of Fatima and no general claims were made regarding the content and distribution of Fatima in wheat genomes. 

6) Line 114, Indicate why these particular chromosomes were chosen for analysis? Why is the location of Ph1 on 5B important to this study? 

Response: Changed (see lines 114-115).

7) Line 123-129. Authors should replace percentages (“5%, 57%, 34% and 4% of the cases”) with actual numbers. 

Response: Done

8) Line 134: How were these 11 cases chosen for study?

 Response: see our response above to editor comment. Explained in lines 128-135. 

9) Line 147: Authors state that “Fatima was most likely activated following allohexaploidization”; the reasoning here is a little murky. Possibly could be helped by summarizing number of copies eliminated. Also, is there any evidence to show that the transposon was inactive before the allohexaploidization?

 Response: It can be deduced from the fact that 80 insertions (~15% of the insertions analyzed here) were found to be wild emmer specific and yet the total number of insertions was similar in wild emmer and bread wheat. Activation fallowing allohexaploidization was previously shown for Fatima as mentioned in lines 80-82.

10) Line 251: DBS, should be DSB 

Response: Done

11) Line 347: templet; should be template?

 Response: Done

12) Line 411, 412: regarding the proposed introgression of an 11 kb segment: Can the TEs contained within this segment indicate its probable origin?

 Response: detail analysis using TREP and in NCBI did not provide any strong hits. We also compered to wild emmer and Ae. tauschii with BLAST+ stand-alone version 2.2.24 and did not find any hit. 

13) Lines 467-469: The potential role of Stasy elements in the segmental duplication is overstated. The first Stasy is 2.5 kb from the duplication endpoint, and the second is indicated as “in the 5’ region” of a 470 kb duplication. As depicted in Figure 4, this second Stasy may be 50 kb or more from the duplication endpoint. If these elements were involved in the duplication events, one would expect them to be present at or very close to the breakpoints. 

Response: We have mentioned that sequence length is unscaled in the Figs. The presence of a truncated Stasy element (SINE) 2.5 kb downstream of the first repeat start in wild emmer and of a highly similar (99%) truncated Stasy element 2.5 kb downstream of the second repeat. Based on Stasy location, we speculated that Stasy might be involved (see lines 404-406). 

14) Line 772: bends, should be bands 

Response: Done

15) Line 775: revers, should be reverse

 Response: Done

16) In this paper, do ‘elimination’ and ‘deletion’ carry the same meaning? 

Response: indeed. And we agree with you, thus we have corrected to “deletion”. 

17) Some loci described in the study have “high” TE content (Lines 186, 214, 230, 263). Considering that wheat genome is already 80% TE sequences, what does “high” mean? 

Response: The TE content is only stated as part of the sequence annotation and characterization, and was not claimed to be high. For clarification, the line discussing the TEs abundant in indels break points was removed from the abstract.

18) It is recommended that Supplemental Figure 7 be included in the manuscript as a main text figure. In fact, Supplemental Figures 4, 5, 6, and 7 could all be combined into a single figure for the main text. These PCR results don’t take much space, and it would be helpful for these data to be readily available to the reader. 

Response: Done. They are now Figs 2 and 6.

---

## [Decision Letter · Decision Letter 1]

19 Feb 2020

PONE-D-19-26784R1

Identification and characterization of large-scale genomic rearrangements during wheat evolution

PLOS ONE

Dear Professor Kashkush,

Thank you for submitting your manuscript to PLOS ONE. After careful consideration, we feel that it has merit but does not fully meet PLOS ONE’s publication criteria as it currently stands. Therefore, we invite you to submit a revised version of the manuscript that addresses the points raised during the review process.

Based on Reviewer 1 and my reading of the manuscript, the following issues need to be addressed.:1. There is a need for further explanation of the logic for using the MITE method to map the Fatima elements.2. Please incorporate each of the experimental, explanatory, and textual comments of Reviewer 1 into the text.3. The labels in Fig 2 are difficult to read. Please use a different font or size.4. Please provide lane designations and other labels for Figure 4.5. It clearer representation of the correlation of  the raw data to the data presented in Figure 2 and Figure 4 is required. Please provide a simple statement in the raw data document of the correspondence of the raw data to the data in the Figures.

We would appreciate receiving your revised manuscript by Apr 04 2020 11:59PM. To enhance the reproducibility of your results, we recommend that if applicable you deposit your laboratory protocols in protocols.io, where a protocol can be assigned its own identifier (DOI) such that it can be cited independently in the future. For instructions see: http://journals.plos.org/plosone/s/submission-guidelines#loc-laboratory-protocols

We look forward to receiving your revised manuscript.

Kind regards,

Arthur J. Lustig, PhD

Academic Editor

PLOS ONE

Reviewers' comments:

Reviewer's Responses to Questions

**Comments to the Author**

1. If the authors have adequately addressed your comments raised in a previous round of review and you feel that this manuscript is now acceptable for publication, you may indicate that here to bypass the “Comments to the Author” section, enter your conflict of interest statement in the “Confidential to Editor” section, and submit your "Accept" recommendation.

Reviewer #1: (No Response)

Reviewer #2: All comments have been addressed

2. Is the manuscript technically sound, and do the data support the conclusions?

Reviewer #1: Yes

Reviewer #2: (No Response)

3. Has the statistical analysis been performed appropriately and rigorously? 

Reviewer #1: N/A

Reviewer #2: (No Response)

4. Have the authors made all data underlying the findings in their manuscript fully available?

Reviewer #1: Yes

Reviewer #2: (No Response)

5. Is the manuscript presented in an intelligible fashion and written in standard English?

Reviewer #1: Yes

Reviewer #2: (No Response)

6. Review Comments to the Author

Reviewer #1: The authors addressed the main concerns and adjusted the format of the manuscript to a more readable format. Despite the the use of a MITE designed tool for the mapping of Gypsy elements that remains unclear to me, the 11 indels presented in the manuscript have been validated and described in detail.

l.11 > might have occurred

l.54 > rephrase to “shaped the wheat genome”?

l.68 > as (was) reported

l.77 > duplicated above (between wheat allopolyploids)

l.91 > structural rearrangements have no “role” in domestication per se, but rather the selection involved during the domestication process will shape the patterns of rearrangements we observe?

l.98 > still unclear how/why MAK was specifically designed to retrieve Fatima insertions. Isn’t the tool designed for MITE elements (inverted repeats)? How is it different from a blast search? or other available tools designed for LTRs?

l.105 > unmapped wording confusing

l.107+113 > reasons confusing to me > rephrase focusing on the fact that the B sub-genome is more variable and bears important genes?

l.109 > remove “(termed the pivotal genome)“

l.118 > Comparative analysis add a note “see Methods part Identification of species-specific Fatima insertions”

l.125 > unclear how this would happen, wasn’t the assemblies used to map the indels and not reads?

l.128-130 > I would clearly state why you’re focusing on those 9 “species specific” insertions

l.138 > “no sequence similarity” to low sequence similarity?

l.157 > “some nucleotide identity“ to sequence homology? Also small paragraph that can be included with the following one?

l.160 > absent sequences to sequences absent

l.172 > coding for a lipoxygenase

l.176 to 180 > methods

l.341-342 > duplicate of l.333-334

l.345 > unclear why “via recurrent backcrossing” here. Also I would tone down the introgression statement as there is no strong evidence for it.

l.360 > “in the sequence coverage” a bit misleading? Rather alignment?

l.379 > source to origin?

l.380 > transpired to emerged?

l.509 and further > complimentary to complementary

l.537 > acting via to promoting?

Reviewer #2: (No Response)

7. PLOS authors have the option to publish the peer review history of their article (what does this mean?). If published, this will include your full peer review and any attached files.

Reviewer #1: No

Reviewer #2: No

---

## [Author Response · Author response to Decision Letter 1]

10 Mar 2020

Dear editor,

We want to thank you for the constructive comments, and especially we want to thank reviewer #1 for his very critical reading of the manuscript. We have addressed all of your minor comments (see below our point by point response to all comments).

Response to editor comments: 

1. There is a need for further explanation of the logic for using the MITE method to map the Fatima elements.

Response: MITE analysis kit (MAK) software (http://labs.csb.utoronto.ca/yang/MAK/) is a homology-based software, meaning it uses a consensus sequence as query and BLASTN algorithm with global alignment. This software was originally developed by Guojun Yang, University of Toronto (Janicki et al. 2011; Yang and Hall 2003a) and devoted for retrieving MITE insertion, so it was named accordingly. However, the MAK can be used for any given query sequence in order to be retrieved from genome draft. We previously used MAK to retrieve retroelements such as Au-SINE [see our previous publications: Ben-David, S., Yaakov, B ., and Kashkush, K. (2013). Genome-wide analysis of short interspersed nuclear elements (SINEs) revealed high sequence conservation, gene association and retrotranspositional activity in wheat. The Plant Journal. 76(2): 201-210; Danielle Keidar, Chen Doron, and Khalil Kashkush (2018). Genome-wide analysis of a recently active retrotransposon, Au SINE, in wheat: content, distribution within subgenomes and chromosomes, and gene associations. Plant Cell Reports. 37(2):193-208]. The big advantage of using the MAK software is that it is very efficient and allows the retrieving of flanking sequences. We have stated the fact that MAK can be used for any TE sequence in lines 290-292. 

2. Please incorporate each of the experimental, explanatory, and textual comments of Reviewer 1 into the text.

Response: See our below response to the reviewer’s comments 

3. The labels in Fig 2 are difficult to read. Please use a different font or size. 

Response: Done. See corrected figure 2.

4. Please provide lane designations and other labels for Figure 4. 

Response: Done. We have corrected the legends of all figures, and included all details. See figure legends. 

5. It clearer representation of the correlation of the raw data to the data presented in Figure 2 and Figure 4 is required. Please provide a simple statement in the raw data document of the correspondence of the raw data to the data in the Figures. 

Response: Done. The raw data refers to figures 2 and 6. Please see the corrected legend in the bottom of “S1_raw_images”. 

Response to Reviewer’s comments:

l.11 > might have occurred. 

Response: Done.

l.54 > rephrase to “shaped the wheat genome”?

Response: Done.

l.68 > as (was) reported. 

Response: Done.

l.77 > duplicated above (between wheat allopolyploids). 

Response: Previous paragraph refers to newly formed wheat allopolyploids, while here we refer to wild relatives and different cultivars. Different articles were brought as references to support the different statements. 

l.91 > structural rearrangements have no “role” in domestication per se, but rather the selection involved during the domestication process will shape the patterns of rearrangements we observe?

Response: we agree with the reviewer and we have erased this sentence. 

l.98 > still unclear how/why MAK was specifically designed to retrieve Fatima insertions. Isn’t the tool designed for MITE elements (inverted repeats)? How is it different from a blast search? or other available tools designed for LTRs?

Response: MITE analysis kit (MAK) software (http://labs.csb.utoronto.ca/yang/MAK/) is a homology-based software, meaning it uses a consensus sequence as query and BLASTN algorithm with global alignment. This software was originally developed by Guojun Yang, University of Toronto (Janicki et al. 2011; Yang and Hall 2003a) and devoted for retrieving MITE insertion, so it was named accordingly. However, the MAK can be used for any given query sequence in order to be retrieved from genome draft. We previously used MAK to retrieve retroelements such as Au-SINE [see out previous publications: Ben-David, S., Yaakov, B ., and Kashkush, K. (2013). Genome-wide analysis of short interspersed nuclear elements (SINEs) revealed high sequence conservation, gene association and retrotranspositional activity in wheat. The Plant Journal. 76(2): 201-210; Danielle Keidar, Chen Doron, and Khalil Kashkush (2018). Genome-wide analysis of a recently active retrotransposon, Au SINE, in wheat: content, distribution within subgenomes and chromosomes, and gene associations. Plant Cell Reports. 37(2):193-208]. The big advantage of using the MAK software is that it is very efficient and allows the retrieving of flanking sequences. We have stated the fact that MAK can be used for any TE sequence in lines 290-292. 

l.105 > unmapped wording confusing 

Response: corrected to “unassigned to a specific chromosome”.

l.107+113 > reasons confusing to me > rephrase focusing on the fact that the B sub-genome is more variable and bears important genes? 

Response: corrected. We have erased this statement. 

l.109 > remove “(termed the pivotal genome)“ 

Response: Done.

l.118 > Comparative analysis add a note “see Methods part Identification of species-specific Fatima insertions”. 

Response: Done. 

l.125 > unclear how this would happen, wasn’t the assemblies used to map the indels and not reads? 

Response: The word “reads” was removed. The insertions might be monomorphic, however, due to Ns in the assembly they were identified as wild emmer unique previous to manual analysis.

l.128-130 > I would clearly state why you’re focusing on those 9 “species specific” insertions.

Response: “The 9 loci together with a single locus out of the 19 previously described loci, showed clear breakpoints and thus were chosen for further analysis.”

l.138 > “no sequence similarity” to low sequence similarity? 

Response: Done.

l.157 > “some nucleotide identity“ to sequence homology? Also small paragraph that can be included with the following one?

Response: We refer to single nucleotide identity in some of the cases. However, we changed this to “sequence homology” as was suggested by the reviewer. The small paragraph refers to 9 different cases, while the following refers only to 3 cases described under the subtitle “Indels flanked by long sequence repeats”. The other 6 cases are discussed under the subtitle “Indels flanked by short sequence repeats”.

l.160 > absent sequences to sequences absent. 

Response: Done.

l.172 > coding for a lipoxygenase. 

Response: Done

l.176 to 180 > methods. 

Response: Done

l.341-342 > duplicate of l.333-334. 

Response: Done

l.345 > unclear why “via recurrent backcrossing” here. Also I would tone down the introgression statement as there is no strong evidence for it. ?

Response: the definition “via recurrent backcrossing” was adapted from [1]. 

The statements regarding introgression are rephrased based on the reviewer 

suggestion. 

l.360 > “in the sequence coverage” a bit misleading? Rather alignment? 

Response: Done

l.379 > source to origin? 

Response: Done

l.380 > transpired to emerged? 

Response: Done

l.509 and further > complimentary to complementary 

Response: Done

l.537 > acting via to promoting? 

Response: Done

---

## [Editor Report · Decision Letter 2]

23 Mar 2020

Identification and characterization of large-scale genomic rearrangements during wheat evolution

PONE-D-19-26784R2

Dear Dr. Kashkush,

We are pleased to inform you that your manuscript has been judged scientifically suitable for publication and will be formally accepted for publication once it complies with all outstanding technical requirements.

With kind regards,

Arthur J. Lustig, PhD

Academic Editor

PLOS ONE

Additional Editor Comments (optional):

All of the reviewer's and editorial comments have been addressed in an appropriate manner.
---

## [Editor Report · Acceptance letter]

27 Mar 2020

PONE-D-19-26784R2 

Identification and characterization of large-scale genomic rearrangements during wheat evolution 

Dear Dr. Kashkush:

I am pleased to inform you that your manuscript has been deemed suitable for publication in PLOS ONE. Congratulations! Your manuscript is now with our production department. 

With kind regards,

on behalf of

Dr. Arthur J. Lustig 

Academic Editor

PLOS ONE